# Genome-Wide Survey and Expression Analyses of Hexokinase Family in Poplar (*Populus trichocarpa*)

**DOI:** 10.3390/plants11152025

**Published:** 2022-08-03

**Authors:** Mei Han, Xianglei Xu, Yuan Xiong, Haikun Wei, Kejun Yao, Tingting Huang, Yingle Long, Tao Su

**Affiliations:** 1Co-Innovation Center for Sustainable Forestry in Southern China, College of Biology and the Environment, Nanjing Forestry University, Nanjing 210037, China; sthanmei@njfu.edu.cn (M.H.); 1736164453@njfu.edu.cn (X.X.); xiongyuan@njfu.edu.cn (Y.X.); weihaikun@njfu.edu.cn (H.W.); yaokejun@njfu.edu.cn (K.Y.); huangtingting2002@163.com (T.H.); 13613033020@163.com (Y.L.); 2Key Laboratory of State Forestry Administration on Subtropical Forest Biodiversity Conservation, Nanjing Forestry University, Nanjing 210037, China

**Keywords:** *Populus*, hexokinase, sucrose metabolism, sugar signaling, stress and defense

## Abstract

Hexokinase (HXK) family proteins exert critical roles in catalyzing hexose phosphorylation, sugar sensing, and modulation of plant growth and stress adaptation. Nevertheless, a large amount remains unknown about the molecular profile of HXK enzymes in *Populus trichocarpa*, a woody model tree species. A genome-wide survey of HXK-encoding genes, including phylogenies, genomic structures, exon/intron organization, chromosomal distribution, and conserved features, was conducted, identifying six putative HXK isogenes (*PtHXK1-6*) in the *Populus* genome. The evolutionary tree demonstrated that 135 homologous HXKs between 17 plant species were categorized into four major subfamilies (type A, B, C, and D), clustering one plastidic (*PtHXK3*) and five mitochondrial *PtHXKs* grouped into type A and B, respectively. The in silico deduction prompted the presence of the conserved sugar-binding core (motif 4), phosphorylation sites (motif 2 and 3), and adenosine-binding domains (motif 7). The transcriptomic sequencing (RNA-seq) and the quantitative real-time PCR (qRT-PCR) assays revealed that three isogenes (*PtHXK2*, *3,* and *6*) were abundantly expressed in leaves, stems, and roots, while others appeared to be dominantly expressed in the reproductive tissues. Under the stress exposure, *PtHXK2* and *6* displayed a significant induction upon the pathogenic fungi (*Fusarium solani*) infection and marked promotions by glucose feeding in roots. In contrast, the *PtHXK3* and *6* are ABA-responsive genes, following a dose-dependent manner. The comprehensive analyses of the genomic patterns and expression profiling provide theoretical clues and lay a foundation for unraveling the physiological and signaling roles underlying the fine-tuned PtHXKs responding to diverse stressors.

## 1. Introduction

Sucrose is the primary form of photosynthetic carbohydrates produced in source leaves. Depending on symplastic and apoplastic pathways, sucrose is translocated and unloaded to sink tissues (e.g., roots and stems), where it is further metabolized into glucose and fructose (hexoses), maintaining cellular metabolism in different compartmentations [1]. In plants, hexoses are the central carbon energy source and signaling molecules influencing the whole life cycle [2]. During the sucrose metabolism, the catalysis of hexose phosphorylation is intermediated by hexokinase (HXK, EC 2.7.1.1) or fructokinase (FRK, EC 2.7.1.4), involving multiple biological processes [3]. HXKs may evolve from the actin fold protein family, sharing a conserved ATP binding site, surrounded by more variable sequences that determine the substrate affinities and other biochemical properties [4]. The phosphorylated hexoses play fundamental functions in various metabolic processes, including intracellular hexose homeostasis and storage of the phosphate group that is transformed into the energy molecules, adenosine diphosphate (ADP) [4,5]. In contrast to FRKs that catalyze fructose phosphorylation into fructose-6-phosphate (F-6P), HXKs can phosphorylate several hexose substrates, including fructose, mannose, and glucose. Remarkably, the latter is converted to glucose-6-phosphate (G-6P), which triggers the release of the stored energy for plant growth and development [6].

HXKs are moonlighting enzymes virtually in all living organisms, from prokaryotes to eukaryotes, modulating sugar metabolism, signaling transduction, and crosstalk with phytohormone pathways [7]. Molecular and biochemical studies have demonstrated that a small multi-gene family encodes HXKs in model plant species [8,9]. In *Arabidopsis*, six HXKs have been characterized, comprising three isoenzymes (AtHXK1-3) that phosphorylate glucose, and the other three (AtHKL1-3) designated as hexokinase-like (HKL) proteins are deficient in catalytic activity [10]. Only three members (*AtHXK1*, *3*, and *AtHKL1*) are verified as glucose sensors, but display different subcellular targets [11]. In rice (*Oryza sativa*), ten HXK isogenes (*OsHXK1*-*10*) have been isolated, whereas the physiological functions appeared to be diversified owing to playing actions as the positive or negative growth regulator [12,13,14]. Moreover, extensive surveys of several HXK families have been reported in *Zea Mays*, *Manihot esculenta*, *Gossypium hirsutum, Phyllostachys dullish*, *Physcomitrella patens*, and *Jatropha curcas*, suggesting the HXK conserved patterns in diverse plant species [15,16,17,18,19,20].

The typical HXKs contain glucose-binding sites and adenosine phosphate-binding domains [1]. In plants, HXKs are commonly classified into two major groups (type A and type B) based on the subcellular targets determined by the N-terminal signal sequences. The type A HXKs (e.g., *AtHXK3*) contain transit peptides, showing a plastidic target, while HXKs in type B are mitochondrial-associated proteins, possessing a conserved hydrophobic membrane anchor domain [8]. While a few HXK isogenes in monocots that identified the localization to the cytoplasm and nucleus are grouped in type C [7,21]. In addition, the HXKs in bryophytes, lycophytes, and gymnosperms likely lack the conserved peptides, thus belonging to type D [19]. The various intracellular localizations in classified HXKs could envision the physiological roles regarding specific metabolic regulation and stress adaptation in plants [1,8].

Emerging reports implicated that HXKs are involved in glycolysis, sugar sensing, and signaling, affecting plant growth and development. Previous research revealed that the *Arabidopsis* transgenic mutants exhibited sugar insensitive and hypersensitive phenotypes in seedlings via up- and downregulation of *AtHXK1* transcript, suggesting that the dual-functional *AtHXK1* acts as a glucose sensor [22]. The HXK’s roles in sugar sensing were evidenced by the ectopic induction of *AtHXK1* in tomatoes (*Solanum lycopersicum*), leading to the retarded growth, reduced photosynthesis efficiency, and accelerated leaf senescence [23]. Exogenous glucose input in *AtHXK1*-overexpressing *Arabidopsis* repressed the leaf expansion, suggesting an AtHXK1-dependent manner, whereas decreases in *AtHXK1* expression prompted a delayed leaf expansion and senescence process [24]. These results indicated that *AtHXK1* played a dual role in glycolysis and sugar sensing for vital metabolic and physiological processes [25]. In contrast, the *AtHKL1*-overexpressing lines displayed phenotypes very similar to those of the *AtHXK1* mutant (*gin2-1*), suggesting that HKL1 was a negative regulator-mediated crosstalk between glucose and phytohormone pathways [26,27,28]. According to a previous report, the *OsHXK5* and 6 are evolutionarily related to *AtHXK1*, playing roles as glucose sensors in rice [14]. Recently, a rice cytosolic hexokinase *OsHXK7* was identified as involved in sugar signaling and metabolism that impacted seed germination in a glycolysis-dependent manner; however, its signaling role was depressed under O_2_-deficient conditions [12]. Nevertheless, in plants and yeast (*Saccharomyces cerevisiae*), the HXK-mediated glucose-sensing functions constantly throughout the life cycle that appears to be independent of catalytic activities in phosphorylating glucose to G-6P [29,30]. Therefore, it is speculated that HXKs are very diversified that some isoforms may be capable of functioning essentially as metabolic regulators, and others may act as glucose sensors [9].

Sugar signaling in regulating plant growth is coordinated with phytohormone signaling pathways [29,31,32]. Recent work in *Arabidopsis* demonstrated that *AtHXK1* was involved in the regulation of the salicylic acid (SA)-dependent programmed cell death (PCD), which is mediated by the alternation of the myo-inositol biosynthesis [33]. In grape (*Vitis vinifera*), the exogenous abscisic acid (ABA) cannot block the glucose-induced repression of sucrose metabolic genes in *CsHXK1*- or *CsHXK2*-silencing mutants that were insensitive to glucose treatment [34]. Ectopic expression of *AtHXK1* in citrus (*Citrus sinensis* × *Poncirus trifoliata*) guard cells reduced the stomatal conductance and transpiration, leading to improved water-use efficiency (WUE) [35]. *AtHXK1* could reduce hydraulic conductance in response to increased glucose levels via controlling aquaporin gene expression, preserving water levels in leaves [36]. Similarly, the constant induction of *OsHXK1* in rice led to rapid leaf senescence and a reduced chlorophyll level, suggesting that *OsHXK1* may modulate glucose homeostasis and reactive oxygen species (ROS) accumulation [37]. This finding supported the fact that HXKs might maintain a steady state of ADP recycling, which affect H_2_O_2_ formation in the mitochondrion [38]. Suppression of *OsHXK10* expression in rice led to the aberrant anther and impaired pollen development [13]. Surprisingly, silencing tomato *SlHXK1* resulted in stunted plant growth and stimulated leaf senescence, associated with an altered starch turnover [39].

HXKs are also involved in stress adaptation and defense regulation in response to various environmental cues and pathogen infection [40,41]. Constantly expressing *AtHXK1* in tobacco (*Nicotiana tabacum*), tomato, and potato (*S. tuberosum*) guard cells increased WUE, conferring tolerance to various abiotic stress (e.g., drought, salt, and heat) [42,43,44]. In a recent report, the different subcellular allocation of serine may be the reason for the retarded growth of the *gin2-1* under high irradiant conditions [45]. It was shown that AtHXK1 acts as a positive regulator of plant immunity in leaves challenged by *Pseudomonas syringae* pv. *tomato* DC3000 through the glucose effect mediated by AtHXK1-related pathways [46]. Overexpression of native *OsHXK1* in rice led to improved ROS accumulation and leaf resistance to virus infection [47]. Moreover, a recent report demonstrated that constant induction of *MdHXK1* expression in apple (*Malus domestica*) enhanced resistance to ring rot fungi pathogen and ROS production related to glucose signaling [48]. In *Populus*, a GATA transcription factor (TF), *PdGNC* was characterized to modulate stomatal aperture and influence WUE and drought tolerance, resulting from activation of hexokinase that promoted NO and H_2_O_2_ accumulation in guard cells [49].

The essential catalytic and signaling roles of HKXs in plant growth and development have been well attempted in *Arabidopsis* and other herbaceous crops. Nevertheless, the molecular conception in genomic patterns and expression profiles, particularly in response to various stress factors, remains largely unknown in forest trees. In this work, we performed a genome-wide survey and expression analyses of the HXKs family in *Populus trichocarpa*, a model woody plant species. The primary objective was to gain novel insights into the molecular aspects of the *HXKs* family in *Populus*. Our data provide a theoretical clue for further unveiling the physiological significance of *PtHXKs* in sucrose metabolism and signaling transduction during plant growth and stress acclimation.

## 2. Results

### 2.1. Genome-Wide Identification and Evolutionary Relationship of HXKs

A systematic pBLAST search using homologs from *Arabidopsis* as queries was conducted in Phytozome v13.1. A total of six putative isogenes were retrieved, proposing to be *PtHXK* after the manual removal of redundant sequences. These gene isoforms were annotated from *PtHXK1* to *PtHXK6* based on the chromosome (Chr) ascending ID number. The gene name, DNA and transcript size, open reading frame (ORF) and protein length, molecular weight (MW), isoelectric point (pI), transit peptide (TP), and subcellular targets are in silico deduced (Appendix A). The sequence length of six *Populus* HXK proteins varied from 494 to 508 amino acid (AA) residues, ranging the MW from 53.09 kDa (PtHXK3) to 54.97 kDa (PtHXK2). All PtHXKs displayed the theoretical acid pI from 5.63 to 6.54. The prediction of intracellular localization indicated that most PtHXKs were localized to the mitochondria, except for PtHXK3, targeting chloroplasts.

The homologous HXKs were identified in 16 other plants, including five monocots: *Brachypodium distachyon*, *O. sativa*, *P. edulis*, *Sorghum bicolor*, and *Z. mays*; eight eudicots: *A. thaliana*, *Eucalyptus grandis*, *Glycine max*, *G. raimondii*, *M. esculenta*, *Medicago truncatula*, *N. tobacum*, and *S. lycopersicum*, alongside one gymnosperm: *Ginkgo biloba*, one bryophyte: *P. patens*, and one lycophyte: *Selaginella moellendorffii*. One hundred and thirty-five homologous HXKs were characterized, varying the numbers from 4 to 14 in all selected plant species (Table 1 and Appendix A). The phylogenetic tree revealed that HXKs were clustered into four major subfamilies: Type A, B, C, and D. Type A was typically composed of HXKs from monocots and eudicots with plastidic targets. Type B contained mitochondrial HXKs that were further clustered into three subgroups: Type B-I, type B-II, and type B-III (Figure 1). Few of HXKs, particularly in eudicots, belong to type B-I. While other mitochondrial HXKs shared the common feature between monocots and eudicots, comprising the type B-II. In contrast, type B-III included mitochondrial HXKs, particularly in type B-III. The cytosolic HXKs in monocots were categorized into the type C subfamily. The remaining HXKs in bryophytes, lycophytes, and gymnosperms were grouped in type D, appearing to have variable subcellular patterns. In Populus, three members (PtHXK1, 4, and 6) were classified into the type B-I subfamily, two PtHXKs (PtHXK2 and 5) in type B-II, and one (PtHXK3) in type C.

### 2.2. Chromosomal Location, Cis-Regulatory Elements, and Genomic Structure of HXKs

Analyses of the chromosomal location revealed that the six *PtHXKs* were mapped on five of the 19 Chrs in *Populus*. Two members (*PtHXK1 and 2*) of *PtHXKs* were located on Chr 1, whereas others showed individual distribution on Chr 5, 7, 9, and 18, respectively (Figure 2a). The evolutionary relationship of *PtHXK2* and *5* may propose the occurrence of segmental duplication events during the genome evolution. Comparative analyses of gene promotors between *Arabidopsis* and *Populus* showed that three potential *cis*-regulatory elements in *PtHXKs* represented the most widely spread elements, including the anaerobic induction (ARE), methyl jasmonate (MeJA) (CGTCA/TGACG), and low temperature (LTR). Within phytohormone-regulated elements, ABA-responsive elements (ABRE) and auxin response elements (TGA/AuxRR-core) were identified in three *PtHXKs*. The gibberellin (GA)-responsive elements (GARE/P-box/TATC-box) showed the presence in two *PtHXKs*. The SA-responsive elements (TCA), MYB TF binding site involved in drought response (MBS), and defense and stress response elements (TC-rich) were distributed in a few members of *PtHXKs*. Thereafter, analyses of genomic patterns revealed a more conserved organization of exon/intron in *Populus*, showing eight introns in all *PtHXKs*, whereas the number of introns varied from 6 to 8 in *Arabidopsis* (Figure 2c). *PtHXK2* and *5* were clustered with the maximal length of the DNA sequence owing to the extended sizes of its introns. The high protein sequence identity to HXK homologs in *Arabidopsis* suggested that PtHXK2, 4, and 5 were postulated to be HKL proteins deficient in catalytic activity (Appendix A and Figure 2b).

### 2.3. Conserved Motifs and Domains in Protein Sequences of PtHXKs

Using the MEME web server, two essential HXK domains (HXK1 and HXK2) were typically present in HXKs between *Arabidopsis* and *Populus* (Figure 3a). A total of ten individual motifs were programmed to vary the length of AA residues from 30 to 50 (Figure 3b). The commonly shared motifs showed a more conserved pattern in the HXK family, except for AtHLK2 and PtHXK4, lacking motif 4 due to the missing specific AA residues. A longer sequence distance between motif 6 and 7 was observed mainly for HXKs (e.g., AtHLK1, 2, PtHXK2, and 5). The multiple sequence alignments were conducted using the functional HXKs in various plants (Figure 4). It was revealed the presence of several conserved fragments, including the N-terminal transmembrane anchor domain (5–24 AA), two phosphate domains (97–118 AA and 248-267 AA), sugar-binding core for substrate recognition (167–186 AA), adenosine phosphate-binding domain (425–461 AA), and C-terminal low complexity domain (477–491 AA). The divergent indel (6–10 AA) within the N-terminal adenosine-binding domain was verified as the critical sequences to distinguish the HXK and HLK subfamilies. These conserved features have been characterized widely in *Arabidopsis*, rice, tobacco, wheat, and *Sorghum* [1]. Two active amino acid residues (Asp104 and Ser177) labeled with stars were identified to possess HXK catalytic activity, demonstrated in rice and *Arabidopsis* [37,50].

### 2.4. Transcriptomics of PtHXKs in Vegetative Tissues and upon Pathogenic Fungi Infection

The spatiotemporal expression patterns of *PtHXKs* were initially analyzed using transcriptomic sequencing (RNA-seq) retrieved from the Phytozome (v13.1). It was demonstrated that a varied transcript abundance was detected in seven selected vegetative tissues, including the root tips (RTP), roots (RT), stem internodes (STI), stem nodes (STN), leaf expanded fully (LFF), leaf immature (LFI), and leaf young (LFY). As depicted in the heat map, *PtHXK2* and *6* were more abundantly expressed in all vegetative tissues, mainly showing the highest levels in the roots and stems (Figure 5a). While *PtHXK3* displayed explicit transcript abundance in the stems, followed by the root tips. The *PtHXK1* and *5* appeared to be not significantly expressed. However, along with *PtHXK4*, their transcripts were predominantly detected in the inflorescence (Appendix A). The expression patterns of six *PtHXKs* in young leaves (YL), mature leaves (ML), stems (ST), and roots (RT) were further evaluated using the in vitro cultured plants by quantitative real-time PCR (qRT-PCR) that was compatible with the RNA-seq data except for *PtHXK5*, showing exceptional levels of low expression (Figure 5b). In addition, the responding *PtHXKs* were investigated in the roots under the time-coursed infection of the pathogenic fungi, *Fusarium solani* (Fs). Taken together, analyses of the tissue-specific expression confirmed that three *PtHXKs* (*PtHXK2*, *3*, and *6*) showed a higher expression level in the roots than other isogenes. These three genes also showed a strong induction upon the Fs infection, particularly after 24 h (h) post-inoculation (hpi) (Figure 5c). The *PtHXK1* transcript appeared to be merely affected before 24 hpi. The time coursed expression evaluated via qRT-PCR assay confirmed that *PtHXK3* and 6 were the Fs responsive genes.

### 2.5. Effects of PtHXKs Expression Responding to Sugars and ABA treatments

The deduced *cis*-regulatory elements on promoters reflect spatiotemporal expressions of specific genes that may be affected by various environmental cues and phytohormone exposures. The gene promoters’ analyses revealed a few ABA-related response elements (Figure 2b). Under the input of different sugars (e.g., glucose, fructose, and sucrose) with gradient levels, the compared expression profile of *PtHXKs* was explored between 0 and 24 h by qRT-PCR assay (Figure 6). The significant inductions of *PtHXK2* and *6* were identified in the roots by adding glucose when the amount increased higher than 3%. In contrast to the control (0 h), the *PtHXK3* appeared to be significantly promoted by 6% sucrose and fructose owing to the observed high expression levels after 24 h of inoculation. In addition, the marked upregulation of *PtHXK6* expression was detected in response to sucrose feeding, following a dose-dependent manner. The HXK-mediated sugar signaling profoundly affects promoting or arresting plant morphogenesis in association with altered ABA levels [8]. In line with this, the effect on *PtHXK* transcripts was inspected in roots upon different dose feeding of ABA. Notably, the constant promotions of *PtHXK3* and *6* were observed to be correlated with the increase in the ABA concentration. Nevertheless, detecting other *PtHXKs* expressions was unsuccessful due to the low transcript abundance in the selected tissues.

## 3. Discussion

### 3.1. The HXKs Family and Conserved Profiles in Plant Species

Recent advances in high-throughput technologies and multi-omics-based strategy have boosted the framework in functional genomics and metabolic research in plants [51]. As a model perennial woody plant species, the entire genome of *Populus* has been released for over 15 years, whereas a thorough survey of the *PtHXKs* family remains unknown. The putative HXK-encoding genes were identified within 17 selected plant species, including 6-14 members (Table 1). Along with other plant species, the phylogenies of 135 *HXK* homologs revealed a classification of four subfamilies (type A, B, C, and D), showing one plastidic *PtHXK3* in type A and five members of mitochondrial *PtHXKs* in type B [1,8]. However, there were no cytosolic HXKs identified in the *Populus* genome. Based on phylogenetic relationships and the protein sequence identity to *Arabidopsis*, three members (*PtHXK2*, *4*, and *5*) most likely belong to HKL-like proteins, while other gene isoforms are HXK candidates in *P. trichocarpa*. The genome duplication facilitates woody perennials to variable environments adaption during million years of evolution [52]. One gene pair (*PtHXK2* and *6*) was speculated to undergo a segmental duplication combined with the sequence analyses, suggesting notable physiological roles associated with evolved mechanisms to adapt to stress stimuli (Figure 2a). The *cis*-regulatory elements predicted in promoter regions indicated the dynamic regulation and evolution of specific gene expression upon various environmental cues [53]. Therefore, the deduction of principal diverse *cis*-regulatory elements related to phytohormone response (ABA, GA, MeJA, and Auxin) and stress factors (e.g., MBS, TC-rich, and LRT), along with TF binding sites, provided a hint that the molecular regulation of *PtHXKs* depends significantly on the crosstalk between phytohormone and glucose signaling pathways [31]. In addition, the analyses of the genomic structure of PtHXKs compared with homologs in *Arabidopsis* suggested a similar exon and intron organization (e.g., eight introns included) within the same subgroup (Figure 2c).

Furthermore, approximately ten conserved motifs (30–50 AA) of HXKs have predicted distribution between *Arabidopsis* and *Populus*, which was in line with the previous reports [1,15,16,34]. The multiple alignments of PtHXK protein sequences with other functional homologs led to the identification of six hallmarks, including 19–50 AA transit peptide sequences (membrane anchor domain) at the N-terminus for the plastidic or mitochondrial targets (Appendix A). The prediction of theoretical core sites for glucose and phosphate binding demonstrated that all homologous HXKs possessed the conserved AA residues (e.g., Asp-101, Gly-104, and Ser-177) except for AtHKL1 and PtHXK4, suggesting that PtHXK4 may be a catalytically inactive glucose sensor (Figure 4). These results were supported by several recent works performed on Cassava, bamboo, and cotton [16,17,18]. These conserved features of HXKs may be more variable in gymnosperms, bryophytes, and lycophytes, suggesting that the plant-derived HXKs probably evolved from prokaryotic ancestors, common primordial actin fold protein [54,55]. Nevertheless, despite the advanced molecular information concerning functional HXKs in various plants, hitherto, no strategy was potentially used to predict the sensor role of a specific HXK, and not all HXKs played identical functions in *Arabidopsis* [8].

Investigating gene expression patterns is one of the strategies to imply whether candidates may be involved in specific metabolic processes or signaling roles. The significant challenge for the tree functional study of a particular gene family is to overpass the relationships between the transcript abundance and corresponding variation of the enzyme kinetics [56]. In our work, the depicted gene expression profiling revealed that a few genes (e.g., *PtHXK2* and *6*) displayed specific expression patterns in vascular tissues, including the roots and stems, reflecting the potential role during plant growth and development. However, the expression of gene pairs (*PtHXK2* and *5*) with segmental duplication was observed to show a divergent pattern (Figure 5). The gene alternative splicing (AS) was prevalent in generating variation in protein structure, functional diversity, and stress adaptation among different plant tissues, cell types, and treatments [57]. Therefore, it was hypothesized that the expression variation of gene duplication might be due to the overlapped levels of transcript variants derived from AS. It is worthwhile to note that when focusing on the effects of *PtHXKs* in the roots responding to the *F. solani* infection, the *PtHXK3* and *6* showed significant increases in expression, particularly at Fs 48 hpi. In contrast to the control, the *PtHXK2* and *6* identified showed a marked promotion in roots upon the gradient glucose (3% and 6%) feeding at 24 h, suggesting the role of the sugar sensor. Only the *PtHXK2* was expressed significantly by adding the highest amount (6%) of fructose. Moreover, the *PtHXK3* and *6* were highly responsive to the sucrose and ABA treatments, suggesting the crosstalk regulation between the sugar and phytohormone under the biotic stress conditions. In contrast, a much lower number of HXK transcripts were significantly influenced in mature leaves than in roots under the above conditions, indicating that this chemical concentration was a significant dosage for roots, but not for aerial leaf tissues within 24 h (data not shown). Overall, transcriptional and post-transcriptional interferences of HXK isogenes in specific tissue types indicate the subsequent exploration of physiological and signaling functions under different stress regimes. Analysis of *cis*-elements and gene expression revealed that *PtHXKs* contained various defense/stress-related response sequences that might be modulated by phytohormone (e.g., ABA), indicating the crosstalk between HXKs and sugar/phytohormone signals during the root development and stress tolerance.

### 3.2. The Regulatory Role of HXKs in Defense and Stress Acclimation

According to the previous research, *Arabidopsis AtHXK1* was identified as a glucose sensor to interrelate nutrient, light, and phytohormone signaling networks for regulating growth and development by responding to various environmental cues [50]. Increasing evidence implicated that HXKs are primary enzymes and exert numerous regulatory actions other than merely catalyzing the phosphorylation of hexoses, the central energy nutrients, and signaling molecules in the effect of sugars on morphogenesis and stress adaptation [7,58]. The glucose acts as a signal molecule through interactions with IAA, GA, and ABA, while the HXK-catalyzed G-6P was regarded as the core intermediate in glucose signal transduction [59]. HXKs with catalytic activity significantly impact multi-cellular processes and sugar signaling, accounting for the metabolite biosynthesis on the glycolytic pathways, providing energy for cell growth [8]. During the plant-pathogen interaction, the host defense and immune responses are mounted with a profound modulation of the primary plant metabolism, including biosynthesis of carbohydrates, amino acids, and the derived secondary metabolites [60].

Feeding experiments elucidated the link between carbohydrate metabolism and defense mechanisms and revealed the induction of pathogenesis-related (PR) genes by sugars, suggesting that carbohydrate metabolism positively regulates the expression of defense-related genes [61]. As an evolutionarily conserved glucose sensor, the *AtHXK1* played dual functions in sugar metabolism, sensing, and defense response [62]. In transgenic tobacco, overexpressing invertase led to sugar accumulation, and the induction of *PR-1* and *PR-5* by glucose in correlation with AtHXK1 mediated the signaling and catalytic activities, suggesting the positive regulation of defense-related genes through carbohydrate metabolism [24]. Constant induction of *AtHXK1* and *2* transcripts in *Arabidopsis* resulted in elevated resistance to the infections of necrotrophic fungi (*Alternaria brassicicola*) and bacterial *P. syringae*, indicating the defense and immunity roles of HXK against various pathogens [40,46]. The underlying mechanisms of HXK involved in regulating immunity and defense appeared to be related to altered ROS accumulation, further supported by several recent works performed on rice and apples [47,48].

Nevertheless, the AtHXK1 might play a negative role as the suppression/deletion of *AtHXK1* led to the increased H_2_O_2_ production associated with defense-related genes and accelerated SA-dependent PCD, prompting that AtHXK1 might exert dual regulatory roles mediating up- and downregulation of sugar responsive genes [33]. The HXK-derived metabolic processes regulate plant defense and immunity to the pathogen, reaching far beyond phytohormone cues and crosstalk between signaling pathways and sucrose metabolism. Recently, two grape *HXK* isoforms and sucrose metabolic enzyme genes increased concomitantly with elevated levels of endogenous glucose and ABA during grape berry development, postulating the regulatory role of glucose and ABA on HXK-dependent sugar metabolism [34]. Emerging reports revealed that ABA signaling pathways regulate water status and mediate drought tolerance by controlling stomatal aperture, water conductance, and gene expression, which were conserved in tree species [63,64]. In line with the previous work, the transcriptional activation of *PdHXK1* via the GATA transcription factor (PdGNC) resulted in the ABA-induced NO and H_2_O_2_ accumulation, mediating the stomatal closure and drought tolerance in woody *Populus* [35,49,65]. In summary, the HXK-derived crosstalk of phytohormone signaling and sugar metabolism deploys defense-related genes and yields profound compounds associated with metabolic conversion, ROS control, and stressor scavenging while remaining a blank in *Populus*.

## 4. Materials and Methods

### 4.1. Plant Materials, Growth Conditions, and Treatments

The *P. trichocarpa* (genotype *Nisqually-1*) grows under long-day conditions (25 °C, 16/8 h day/night photoperiod, 50 µE), culturing in vitro on a traditional woody plant medium (WPM) with 30 g L^−1^ sucrose, 0.1 mg L^−1^ IBA, and solidified with 8 g L^−1^ plant agar. The Fs culture, the fungal spore calculation (1.0 × 10^6^ spores/mL), and the root infection were conducted according to the report [66]. For feeding experiments, the 4-week in vitro cultured plants were transferred to conical flasks containing 40 mL gradient concentrations (0%, 1%, 3%, and 6%) of solution and incubated in an artificial chamber for 24 h. For RNA-seq data, different vegetative tissues and roots with time-course (0, 24, 48, and 72 hpi) of Fs infection were sampled based on the previous research [67]. The roots of eight randomly selected plants with Fs infection and 24-h-feeding by sugars (glucose, fructose, and sucrose) and ABA (0, 50, 100, and 200 µM) were pooled for one biological replication to evaluate gene expressions using the 0 h treatment as the control.

### 4.2. Sequence Mining, Identification, and Genomic Analyses of HXKs

The *Arabidopsis* HXK homologs were collected in TAIR. Available online: https://www.arabidopsis.org (accessed on 24 October 2000) as queries to search for candidate isogenes in *P. trichocarpa* genome assembly (v3.1) from the JGI gene catalog. Available online: https://phytozome-next.jgi.doe.gov/info/Ptrichocarpa_v3_1 (accessed on 30 November 2018) with the E-value cutoff set as 1e-5. Incomplete protein sequences with short lengths (<300 AA) were eliminated. The upstream 1.5 kb sequences of gene promotors were predicted by the program PlantCARE. Available online: http://bioinformatics.psb.ugent.be/webtools/plantcare/html (accessed on 11 September 2000) to obtain overviews of *cis*-regulatory elements associated with the responsiveness of biotic and abiotic stresses, according to the previous reports [60]. The gene structure was deducted by comparing CDS and the corresponding DNA sequence in GSDS. Available online: http://gsds.gao-lab.org (accessed on 1 January 2015). The chromosomal distribution of *PtHXKs* was obtained from the PopGenIE. Available online: http://popgenie.org/chromosome-diagram (accessed on 1 January 2021), and physical locations were drawn by the program MapInspect. Available online: http://www.softsea.com/review/MapInspect.html (accessed on 9 November 2010) [68]. Subcellular localization of PtHXKs was predicted by the program DeepLoc 2.0. Available online: https://services.healthtech.dtu.dk/service.php?DeepLoc-2.0 (accessed on 22 March 2021). Protein sequences of HXKs in *B. distachyon*, *S. bicolor*, *E. grandis*, *G. max*, and *M. truncatula* are available from the Phytozome v13. Available online: https://phytozome-next.jgi.doe.gov (accessed on 28 July 2021) and homologs of *G. biloba* are retrieved from the database. Available online: https://ginkgo.zju.edu.cn/genome (accessed on 1 January 2022). Other protein sequences in *O. sativa*, *P*. *edulis*, *Z*. *mays*, *A. thaliana*, *G. raimondii*, *M. esculenta*, *N. tobacum*, *S. lycopersicum*, *P. patens*, and *S. moellendorffii* were obtained from the literatures [1,10,15,16,17,18].

### 4.3. Phylogenetic Tree Construction and Analyses of Conserved Motifs

The alignments of multiple protein sequences were conducted by Clustal Omega. Available online: https://www.ebi.ac.uk/Tools/msa/clustalo (accessed on 23 May 2018). The reciprocal pBLAST was conducted to establish the genetic relationship between gene pairs. The phylogenetic tree was constructed by MEGA X. Available online: https://www.megasoftware.net/v10.2.2 (accessed on 1 October 2020), using the Maximum Likelihood method with 1000 bootstrap replicates [69]. The evolutionary distances were computed using the Poisson correction method and the number of AA substitutions per site. The conserved motifs were analyzed by MEME. Available online: http://meme-suite.org/index.html (accessed on 1 August 2022), setting the maximum numbers and widths of motifs to 10 and 50, respectively [70]. Transmembrane region and low complexity were predicted in the SMATE database. Available online: https://smart.embl-heidelberg.de (accessed on 26 October 2020) based on AtHXK1. The motif was annotated by CDD in NCBI. Available online: https://www.ncbi.nlm.nih.gov/cdd (accessed on 5 November 2021) and ScanProsite. Available online: http://prosite.expasy.org/scanprosite (accessed on 25 May 2022) [71].

### 4.4. Transcriptome and Expression Validation by qRT-PCR

Transcriptome and data processing were performed on the Phytozome (v13.1) and based on a previous report [67]. Gene transcript levels in various tissues were valued by fragments per kilobase of exon model per million mapped reads (FPKM). For Fs infection, the significance of differentially expressed genes (DEGs) (FPKM > 5) was judged by the *p <* 0.05. Both Fisher’s exact test (*p <* 0.05) and multi-test adjustment (false discovery rate (FDR) < 0.05) were applied in DEGs identification based on the report [67]. For the qRT-PCR assay, the RNA extraction and cDNA synthesis were performed, according to the previous report [72]. The primer amplification efficiency was evaluated with dilutions of cDNA, producing an R^2^ ≥ 0.99. The cDNA samples were loaded to a TB green Premix ExTap™ Tli RNaseH Plus (Takara, China). The mixture was subjected to StepOnePlus™ Real-Time PCR System (AB, USA). The relative gene expression was normalized by the geometric mean of three housekeeping genes (*Ptβ-Actin, PtUBIC,* and *PtEF-1α*). The primers used for targeting specific genes are listed in Appendix A. Heatmaps were constructed by the program CIMminer. Available online: http://discover.nci.nih.gov/cimminer/home.do (accessed on 19 July 2018).

## 5. Conclusions

HXKs are multifaceted enzymes, playing essential roles in various metabolic processes and signaling that significantly impact the whole plant cycle, including regulating vegetative growth and reproduction, male fertility, and senescent signals. Despite the substantial evidence of the regulatory role of HXK in plant defense and stress adaptation, still more studies are needed to integrate molecular information on the biochemical properties, cellular localization, and sensor capacities with other sugar-sensing pathways to improve performance in woody plants. Among the six putative *PtHXKs* identified in *Populus*, three isogenes (*PtHXK2*, *3*, and *6*) showed predominant expressions in the vascular tissues (e.g., roots and stems). *PtHXK3* and *6* were significantly induced upon the sugar and ABA treatment, suggesting potential in vivo activities for catalyzing the hexose phosphorylation and signaling effects on *Populus* growth and development. Moreover, the genomic characterization of HXK families is primary for the fine-tuning of HXK-dependent pathways by engineering the activities or expressions of critical HXKs, which might be sufficient to achieve pathogen resistance and abiotic stress tolerance without compromising plant biomass. Therefore, the inspected *PtHXKs* with dominant responsive features to selected stimuli will be the modifying target for functional analyses under stress exposure in *Populus*.

## Figures and Tables

**Figure 1 plants-11-02025-f001:**
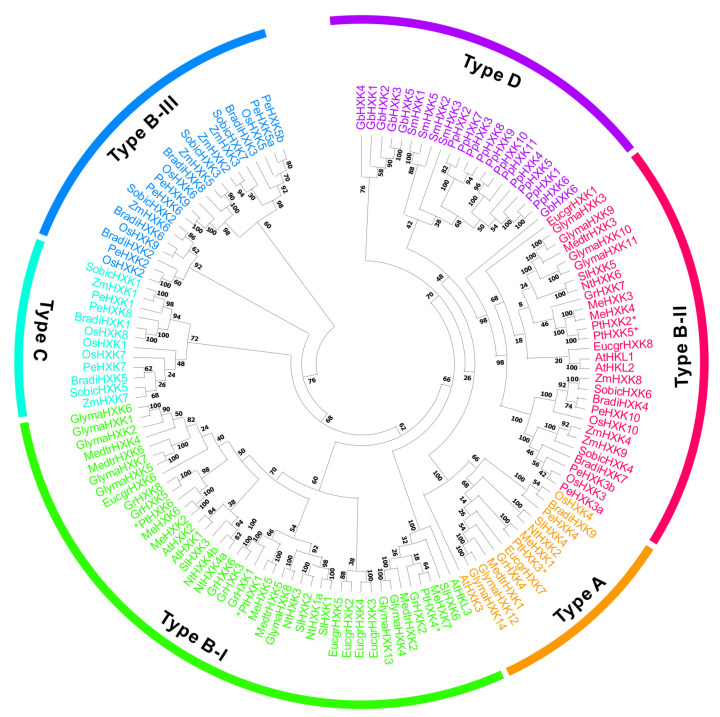
Phylogenetic relationships of clustered homologous HXKs between 17 plant species. The evolution tree was generated using the Maximum Likelihood method in MEGA X with 1000 bootstrap replicates. The four classified HXKs subgroups were represented as type A, B, C, and D.

**Figure 2 plants-11-02025-f002:**
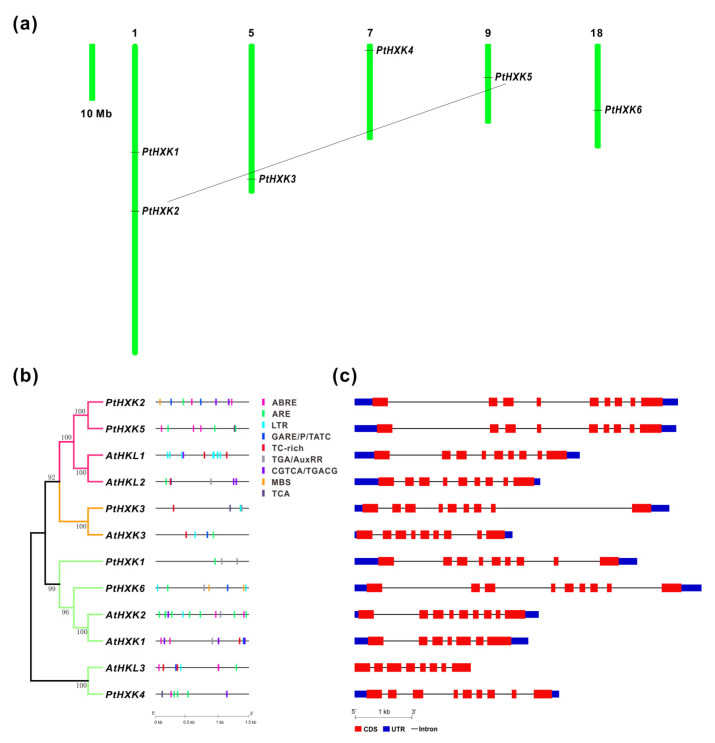
The genomic features of the HXKs between *P. trichocarpa* and *Arabidopsis*. (**a**) The chromosomal distribution of isogenes shows the duplicated gene pair with a straight line. (**b**) The in silico prediction of the *cis*-regulatory elements in 1.5 kb gene promoters labeled in colors. (**c**) Genomic structures show the patterns of the exon/intron organization. The exon lengths are displayed proportionally to the scale on the bottom.

**Figure 3 plants-11-02025-f003:**
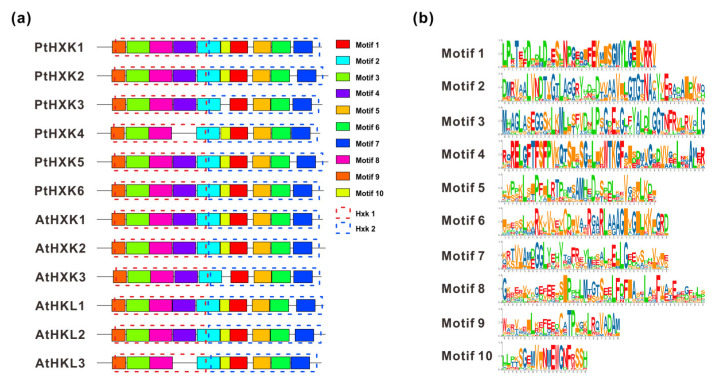
The conserved motif distribution in HXKs. (**a**) Motifs were analyzed using the MEME web server, and ten conserved motifs were boxed in colors. (**b**) The amino acid logos of respective motifs were detailed on the right side.

**Figure 4 plants-11-02025-f004:**
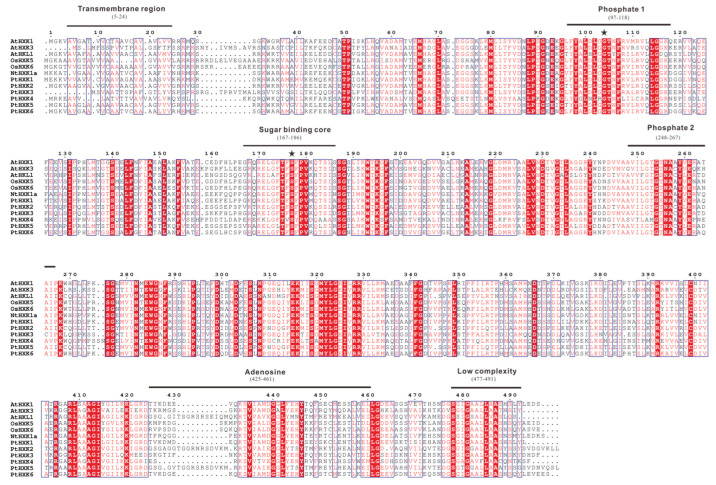
The conserved fragments and amino acid residues between PtHXKs and other functional homologs. Six conserved sites with different numbers of amino acids were lined in black annotated based on homologous regions to the HXK II in yeast, and two active binding sites were marked with asterisks.

**Figure 5 plants-11-02025-f005:**
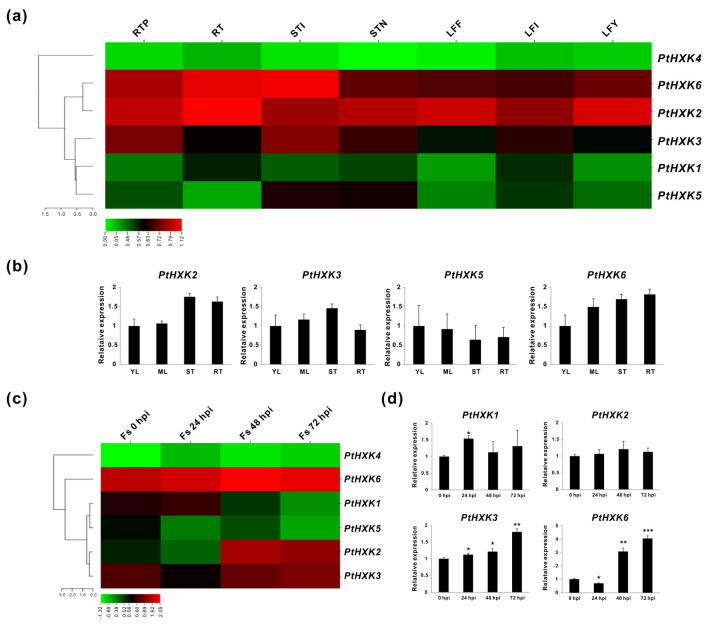
Transcriptomic and expression profiles of *PtHXKs* in vegetative tissues and roots challenged by *F. solani* (Fs). (**a**) The heatmap shows the transcript in vegetative tissues of *P. trichocarpa*. (**b**) The qRT-PCR evaluation shows the tissue-specific expression of *PtHXKs*. (**c**) The heatmap shows the transcriptomic profile of PtHXK isogenes in Fs infected roots (0, 24, 48, and 72 h of post-inoculation, hpi). (**d**) Validation of responsive *PtHXKs* compared to the control (0 hpi) by qRT-PCR. The RNA-seq data were given in the Log10 of the fragments per kilobase per million reads (FPKM) expression values. At least three independent biological replicates were conducted for qRT-PCR analyses. *PtβActin*, *PtUBIC*, and *PtEF-α1* were used as the internal control. Data represent mean values standard error (±SE) of at least three independent biological replicates. Asterisks indicate significant differences relative to the control using the Student’s *t*-test: *** *p <* 0.001, ** *p* < 0.01, and * *p <* 0.05.

**Figure 6 plants-11-02025-f006:**
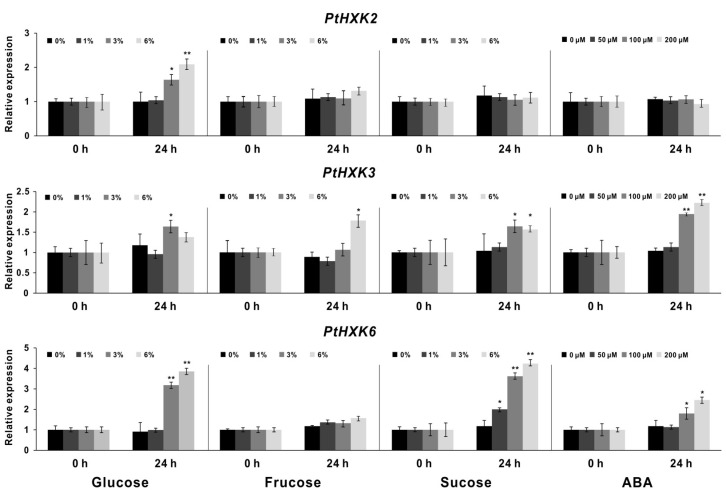
Expression effects of three *PtHXKs* in the roots feeding with various sugars and ABA. The qRT-PCR analyses show the transcript responsiveness upon the altered concentration of sugars (glucose, fructose, and sucrose) and ABA. The relative expression of the control was set as 1. Data represent mean values ±SE of at least three independent biological replicates. *PtβActin*, *PtUBIC*, and *PtEF-α1* were used as reference genes. Asterisks indicate significant differences in comparison with the control using the Student’s *t*-test: ** *p <* 0.001, * *p* < 0.01.

**Table 1 plants-11-02025-t001:** List of the classification and numbers of HXKs in various plant species.

Groups	Species	Type A	Type B			Type C	Type D	Total
			I	II	III			
Bryophytes	*Physcomitrella patens*						11	11
Lycophytes	*Selaginella moellendorffii*						4	4
Gymnosperms	*Ginkgo biloba*						6	6
Monocots	*Brachypodium distachyon*	1		2	4	2		9
	*Oryza sativa*	1		2	4	3		10
	*Phyllostachys edulis*	1		3	5	3		12
	*Sorghum bicolor*			2	3	2		7
	*Zea mays*			3	4	2		9
Eudicots	*Arabidopsis thaliana*	1	3	2				6
	*Eucalyptus grandis*	1	5	2				8
	*Glycine max*	2	8	4				14
	*Gossypium raimondii*	1	6	1				8
	*Manihot esculenta*	1	4	2				7
	*Medicago truncatula*	1	4	1				6
	*Nicotiana tabacum*	1	4	1				6
	*Populus trichocarpa*	1	3	2				6
	*Solanum lycopersicum*	1	4	1				6

## Data Availability

The raw sequence reads of RNA-seq were deposited in the NCBI database with the accession BioProject of PRJNA680933 and the accession BioSample, SAMN16927537, including twelve accession numbers of SRR13347970-981 for triplicate data of each *F. solani* treatment (Fs0, Fs24, Fs48, and Fs72).

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
