# Peer review of "Genome-Wide Survey and Expression Analyses of Hexokinase Family in Poplar (Populus trichocarpa)"

_plants, 2022, doi:10.3390/plants11152025_

Round 1

Reviewer 1 Report

In this manuscript, the authors conducted a genome-wide survey of HXK family genes comprehensively, including phylogenies, genomic structures, exon/intron organization, chromosomal distribution, and conserved features. Moreover, the expression pattern of HXK family genes was profiled in the different tissues under Fungi infection and sugar/ABA treatments by RNA-Seq and RT-qPCR. Three PtHXKs displayed predominant expressions in the vascular tissues, and two out of them were significantly induced with sugar and ABA treatments, which indicates they may exert critical roles in sugar metabolism and stress response. Such a focused study is of significant for further unveiling the positive roles of key PtHXKs in plant growth and stress acclimation in poplars. However, some small issues should be addressed before it becomes acceptable in Plants.

1.       The authors can add the origin of HXK family genes in the Introduction part.

2.       In the abstract, the authors described that three isogenes (PtHXK2, 3, and 6) were abundantly expressed in leaves, stems, and roots, while others appeared to be dominantly expressed in the reproductive tissues. However, there is no expression analysis of PtHXKs in the reproductive tissues in the study. I'm a litter confused about the conclusion.

3.       The image resolution of Fig. 4 is too low in the review version. The fonts and marks on the pictures are even not clear.

4.       In Fig. 5, what’s meaning of YL, ML, ST and RT? All the English abbreviations should be explained in the figure legends.  

5.       The RNA-Seq data should be submitted to public database such as NCBI SRA with accession number, and make it public upon acceptance to make the study reproducible.

Author Response

In this manuscript, the authors conducted a genome-wide survey of HXK family genes comprehensively, including phylogenies, genomic structures, exon/intron organization, chromosomal distribution, and conserved features. Moreover, the expression pattern of HXK family genes was profiled in the different tissues under Fungi infection and sugar/ABA treatments by RNA-Seq and RT-qPCR. Three PtHXKs displayed predominant expressions in the vascular tissues, and two out of them were significantly induced with sugar and ABA treatments, which indicates they may exert critical roles in sugar metabolism and stress response. Such a focused study is of significant for further unveiling the positive roles of key PtHXKs in plant growth and stress acclimation in poplars. However, some small issues should be addressed before it becomes acceptable in Plants.

1. The authors can add the origin of HXK family genes in the Introduction part.

R1: Thanks for the comments. Gene isoforms that encode HXKs are commonly present in every domain of life, ranging from bacteria, yeast, and plants to humans and other vertebrates. HXKs are categorized as actin fold proteins, sharing a conserved ATP binding site core, surrounded by more variable sequences that determine substrate affinities and properties. Please check the updated ms in Line 42-45 (PDF version).

2. In the abstract, the authors described that three isogenes (PtHXK2, 3, and 6) were abundantly expressed in leaves, stems, and roots, while others appeared to be dominantly expressed in the reproductive tissues. However, there is no expression analysis of PtHXKs in the reproductive tissues in the study. I'm a litter confused about the conclusion.

R2: This comment is quite interesting. The detailed expressions of PtHXKs in the reproductive tissues are not the core structure and critical issue in the manuscript, particularly for the woody Populus. Nevertheless, Figure S1 in the updated supplementary materials explains the concern.

3. The image resolution of Fig. 4 is too low in the review version. The fonts and marks on the pictures are even not clear.

R3: The comment is good. The images with high resolution (>800 dpi) were provided in the updated MS. However, the transformed MS in PDF format may significantly reduce the image quality. Also, please manually set up the word app as "do not compress the image in file" in Options/Advanced when checking the MS in Word format.

4. In Fig. 5, what's meaning of YL, ML, ST and RT? All the English abbreviations should be explained in the figure legends.  

R4: Thanks for the comments. The explanation of the abbreviated letters corresponding to four selected tissues was provided in the updated MS in Line 250-252.

5. The RNA-Seq data should be submitted to public database such as NCBI SRA with accession number, and make it public upon acceptance to make the study reproducible.

R5: Many thanks! Please check the updated MS with accession numbers available for the raw sequence reads of RNA-seq deposited in the NCBI.

Reviewer 2 Report

Hexokinases (HXK)participate in the process of plant respiration and metabolism, and play a dual role in plant sugar sense and sugar signal transduction. The present study identified six hexokinases from Poplar and reported their genomic structure, constitution, conserved domains and expression patterns in response to biotic stress and hormone treatments. The manuscript is overall well written. The introduction and discussion are well documented. My major concern are the results which provide less information for the readers. The aim of the work is not clearly established. The quality of the Figures should be improved for high resolution.

 Followings are some minor concerns for the readers to consider.

 (1) It seems all three reported HXKs in Figure 6 showed enhanced expression in response to sucrose feeding following a dose-dependent manner (not upregulated in low concentration, while upregulated in relatively high concentration).

(2) Describe the abbreviations used in Figure 5

(3) What the log value for FPKM, e.g., log2?

(4) Figure 1, what the asterisk stands for? Cite Table S3 in figure legend for lists of gene ID and name used in Phylogenetic tree

(5) The collinearity relationship between PtHXK2 and PtHXK5 should be based on the alignment of locally segmental sequence rather than a single gene.

Author Response

Hexokinases (HXK)participate in the process of plant respiration and metabolism, and play a dual role in plant sugar sense and sugar signal transduction. The present study identified six hexokinases from Poplar and reported their genomic structure, constitution, conserved domains and expression patterns in response to biotic stress and hormone treatments. The manuscript is overall well written. The introduction and discussion are well documented. My major concern are the results which provide less information for the readers. The aim of the work is not clearly established. The quality of the Figures should be improved for high resolution.

 Followings are some minor concerns for the readers to consider.

(1) It seems all three reported HXKs in Figure 6 showed enhanced expression in response to sucrose feeding following a dose-dependent manner (not upregulated in low concentration, while upregulated in relatively high concentration).

R1: Thanks for the comments. While this makes no sense, in Figure 6, upon the sucrose feeding, only PtHXK6 expression appeared constantly induced, which might follow a dose-dependent manner based on the qRT-PCR. This phenomenon was not identically observed for PtHXK2 and PtHXK3.

(2) Describe the abbreviations used in Figure 5

R2: Many thanks! The updated MS in Line 250-252 (PDF version) explained the abbreviated letters corresponding to four selected tissues.

(3) What the log value for FPKM, e.g., log2?

R3: Thanks for the excellent comment. Log10 has been provided in the updated MS.

(4) Figure 1, what the asterisk stands for? Cite Table S3 in figure legend for lists of gene ID and name used in Phylogenetic tree

R4: This comment is quite interesting. If the MS went through thoroughly, these concerns could be worked out appropriately. The asterisk indicates two active binding sites described in the legend of Figure 4. Table S3 has been inserted behind the sentence "One hundred and thirty-five homologous HXKs were characterized, varying the numbers from 4 to 14 in all selected plant species" in Line 164.

(5) The collinearity relationship between PtHXK2 and PtHXK5 should be based on the alignment of locally segmental sequence rather than a single gene.

R5: This comment is perfect. PtHXK2 and PtHXK5 are the only duplicated gene pair based on the sequence identity and phylogenetic relationship. This description has been revised in Line 186.